

# Improving sustainable use of genetic resources in biodiversity archives

E. J. Tuschhoff[1,2,3], Carl R. Hutter[2,3] and Richard E. Glor[2,3]

[1] Department of Ecology and Evolutionary Biology, University of Arizona, Tucson, AZ, USA
[2] Department of Ecology and Evolutionary Biology, University of Kansas, Lawrence, KS, USA
[3] University of Kansas Biodiversity Institute, Lawrence, KS, USA

## ABSTRACT

Tissue sample databases housed in biodiversity archives represent a vast trove of genetic resources, and these tissues are often destructively subsampled and provided to researchers for DNA extractions and subsequent sequencing. While obtaining a sufficient quantity of DNA for downstream applications is vital for these researchers, it is also important to preserve tissue resources for future use given that the original material is destructively and consumptively sampled with each use. It is therefore necessary to develop standardized tissue subsampling and loaning procedures to ensure that tissues are being used efficiently. In this study, we specifically focus on the efficiency of DNA extraction methods by using anuran liver and muscle tissues maintained at a biodiversity archive. We conducted a series of experiments to test whether current practices involving coarse visual assessments of tissue size are effective, how tissue mass correlates with DNA yield and concentration, and whether the amount of DNA recovered is correlated with sample age. We found that tissue samples between 2 and 8 mg resulted in the most efficient extractions, with tissues at the lower end of this range providing more DNA per unit mass and tissues at the higher end of this range providing more total DNA. Additionally, we found no correlation between tissue age and DNA yield. Because we find that even very small tissue subsamples tend to yield far more DNA than is required by researchers for modern sequencing applications (including whole genome shotgun sequencing), we recommend that biodiversity archives consider dramatically improving sustainable use of their archived material by providing researchers with set quantities of extracted DNA rather than with the subsampled tissues themselves.

## INTRODUCTION

Genetic resources archived in biodiversity collections are critically important for scientific research because they permit immediate access to large numbers of samples obtained across taxa, time and space, including samples that would be difficult or even impossible to obtain today (*Droege et al., 2014*; *Burrell, Disotell & Bergey, 2015*; *Schäffer et al., 2017*). Increasing reliance on archived genetic resources by a growing community of researchers, however, presents a significant challenge because current methods for sharing genetic resources are not sustainable; in most cases, researchers requesting access to genetic

Corresponding author
E. J. Tuschhoff,
etuschhoff@email.arizona.edu

resources are provided with a piece of tissue that is consumptively subsampled from a permanently archived resource (*Zimkus & Ford, 2014*). Researchers then destroy this subsample during the course of DNA extraction, use the DNA that is required for their research and typically discard any remaining material. As a result, every request to use genetic resources results in depletion of samples that, left unchecked, will result in complete sample exhaustion and permanent loss of an irreplaceable resource. Because some tissues are present in very small quantities, some genetic resources can only be provided to one or a few researchers before an irreplaceable resource is lost forever. This issue becomes especially pressing when one considers the current extinction crises and increasingly strict regulations for scientific collecting that may prevent samples being replenished from wild specimens (*Stuart et al., 2004*; *Watanabe, 2015*). As a result, it is important to develop protocols that improve sustainable use of these resources.

Because the vast majority of requests to use archived genetic resources involve efforts to sequence DNA, protocols for DNA extraction from archival tissues are an obvious focal point for optimization aimed at improving sustainability of current practices. Most biodiversity collections aim to provide researchers requesting access to genetic material with enough tissue to conduct two DNA extractions (*Zimkus & Ford, 2014*), but collections staff and researchers are often unaware of how much tissue is optimal for extraction because few studies have investigated how sample age, preservation method, extraction protocol, type of tissue, and subsample size are related to the quantity, concentration, and quality of extracted DNA (but see *Reineke, Karlovsky & Zebitz, 1998*; *Drabkova, Kirschner & Vlcek, 2002*; *Guo et al., 2009*; *Sawyer et al., 2012*; *Choi, Lee & Shipunov, 2015*; *Schiebelhut et al., 2016*; *Abdel-Latif & Osman, 2017*). Even parameters that are known to impact extraction success are rarely quantified when biodiversity collections fulfill requests for access to genetic material. For example, tissue mass is known to be strongly correlated with extraction success (*Hykin, Bi & McGuire, 2015*) and has been shown to be correlated with extracted DNA concentration (*Reineke, Karlovsky & Zebitz, 1998*; *Choi, Lee & Shipunov, 2015*) but collections staff and researchers generally use a coarse visual estimate when removing tissue subsamples and rarely obtain quantitative size or mass data. It is not currently common practice to standardize tissue mass prior to DNA extractions (*Wilcox et al., 2002*; *Aguirre-Peñafiel et al., 2014*; *Naccarato, Dejarnette & Allman, 2015*) or to report masses if they were standardized (*Kayes et al., 2013*) except in experiments to compare various protocols or methods (*Drabkova, Kirschner & Vlcek, 2002*; *Guo et al., 2009*; *Abdel-Latif & Osman, 2017*; *Yalçınkaya et al., 2017*). In publications, researchers tend to qualitatively report the amount of starting material with phrases such as "two small pieces" or "usually minute" (*Hajibabaei et al., 2005*; *Jaksch et al., 2016*).

The goal of the present study is to develop guidelines for more sustainable use of genetic resources in biodiversity collections, with a focus on determining the optimal amount of tissue for DNA extraction from amphibian tissue samples. In our first experiment we test whether the type of coarse visual estimates of tissue mass or size that are used by most collections staff who fulfill requests for access to genetic resources are capable of consistently yielding sufficient DNA for modern downstream sequencing applications. In our second experiment, we identify the tissue masses that result in the most efficient use

of archived samples by conducting controlled extractions across a range of samples with known masses. In our third experiment, we test consistency of extraction success across replicate subsamples of a mass that appears to optimize yield while minimizing depletion of the archived samples during a single extraction. In our fourth and final experiment, we test whether our protocol is suitable for samples archived over a 25-year interval from 1984 (around the time collections started accumulating sample preserved specifically for use in molecular genetic studies) until 2001. Given the nature of natural history collections, it is probable that researchers will need to work with tissues of a variety of ages. Previous studies of bone and plant tissues have not recovered a significant correlation between DNA yield and tissue age (*Sawyer et al., 2012*; *Choi, Lee & Shipunov, 2015*), and, to our knowledge, previous published studies have not tested the correlation between age and total DNA yields using cryogenically preserved soft tissues from vertebrates. However, one study of herpetological specimens found a significant decrease in recovered sequence length as tissue age increased (*Chambers & Hebert, 2016*).

## MATERIALS AND METHODS

### Sampling

We conducted all our experiments on amphibian tissues samples from the herpetological collection at the University of Kansas Biodiversity Institute. With more than 40,000 tissue samples in cryogenic storage, this collection is among the largest archives of its kind. This collection is also widely used by the scientific community, with more than 75 requests for access to genetic resources by the scientific community resulting in subsampling of more than 1,100 archived samples over the past 5 years. We focused on liver and muscle tissue because these tissues are the most abundant in biodiversity archives and are usually the standard tissue types collected in the field. Tissues were initially preserved using one of two strategies: immersion in high concentration ethanol or flash freezing in liquid nitrogen. Subsequent to initial preservation, samples were stored in a cryogenic facility, either in mechanical ultra-cold freezers at −80 °C (experiments 1–3) or liquid nitrogen cooled dewars at −180 °C (experiment 4).

### Tissue extraction protocol

The majority of the tissues used in this experiment were stored in ethanol solution. Tissues that had been flash frozen and were not stored in ethanol solution were transferred to a 95% ethanol solution and allowed to thaw to −80 °C such that all tissues were under the same conditions at the time of massing. All tissues were next removed from ethanol and the ethanol was allowed to evaporate for up to 2 min to limit the contribution of ethanol to inferred tissue mass. Each tissue was subsampled with a sterile razor blade until the mass was within 0.5 mg of the target mass as measured by a Mettler Toledo XS105DU analytical balance scale (in eight cases, masses more than 0.5 mg under the target mass were used because there was not adequate tissue remaining for the full amount, see additional details below). Tissues were then placed in a solution of 10 μL protein kinase and 190 μL lysis buffer and incubated at 55 °C for approximately 24 h (several of the larger masses required longer incubation times for complete tissue digestion as determined by

the absence of solid tissue pieces in the solution). Tissue solutions were vortexed once at the start of the incubation period for 10 s and one to three times at the end of the incubation period depending on the level of tissue digestion.

The extractions in this experiment were performed using the Promega Maxwell RSC Instrument (Promega Corporation, Madison, WI, USA). The Maxwell RSC uses paramagnetic particles along with magnetic plungers to lyse and capture DNA along with specialized reagents provided in single use cartridges (*Kephart et al., 2006*). Aside from lysis and transfer to a sterile Eppendorf tube for quantification and storage, the extraction process is entirely automated and occurs inside the instrument. This method was chosen for our experiments for three reasons, and in spite of the fact that the method has relatively high costs both in terms of initial investment in the machine (>$20,000) and for individual extractions (~$8 per cartridge) as of June 26, 2019. First, a recent comparative analysis of commonly used extraction protocols found that the Promega paramagnetic particle method results in particularly high DNA yields, high sample efficacy (measured in the success of PCR), and low error (*Schiebelhut et al., 2016*). Secondly, this automated extraction method allows for a high degree of uniformity across multiple trials and reduces the human error inherent in manual protocols. Finally, third, the Promega RSC instrument relies on sterile individual use cartridges, a drip-free protocol, and includes an automated UV sterilization of internal components following each extraction, which collectively minimize the potential for contamination.

In our study, we used the Promega blood DNA purification kit (Promega product ID: AS1010). We followed the manufacturer's procedures (Maxwell(R) RSC Blood DNA Kit technical manual TM419; Promega, Madison, WI, USA) during the extraction except that elution buffer volume was doubled to 100 µL because at lower volumes the quantity of DNA could not be read by a fluorometer as DNA concentrations were too high. After extraction was completed, quantifications were performed using a Promega Quantus fluorometer.

## Experiment 1: Testing the effectiveness of the "eyeball" method for obtaining tissues appropriate for extraction

We first conducted a preliminary experiment to determine if coarse visual assessment of tissue mass (i.e., the "eyeball" approach to tissue quantification used by most biodiversity collections staff) is capable of sampling tissues that result in consistent DNA yield which are sufficient for modern downstream DNA sequencing applications. The concentration and amount of DNA required for sequencing depends on the sequencing method used, ranging from less than 10 ng of DNA for Sanger sequencing a single DNA fragment to 500 ng for Illumina Truseq-style library preparation (*Hutter et al., 2019*) to over 1,000 ng for high coverage sequencing of an entire vertebrate genome via the Illumina platform (*Arbor Biosciences, 2019*). Because 1,000 ng is at the high end of the amount used for standard sequencing methods applied to typical vertebrate genomes (including whole genome sequencing and popular methods such as RADseq and probe capture), we used this amount as our threshold for establishing extraction success.

For this experiment, two experienced scientists (Drs. Carl Hutter and Shea Lambert) attempted to consistently subsample tissues with a mass considered sufficiently large for

DNA extraction based on prior experience. Tissue subsamples obtained in this manner were then weighed prior to extraction and quantification. Although the researchers knew that their subsamples were being massed, they were asked to subsample per their normal procedures and were not given any feedback about the masses of their samples. Following extraction, we tested whether each sample passed our 1,000 ng minimum threshold for successful extraction. We also tested the basic prediction that tissue mass is correlated with DNA yield using a Pearson's correlation test. Finally, we tested reliability of "eyeball" estimates of tissue mass by estimating variance in both the mass and DNA yield of resulting subsamples.

## Experiment 2: Identification of optimal tissue mass for effective and efficient extraction

Our second experiment focused on identifying the optimal tissue masses for DNA extraction, which we define here as the masses that results in high DNA yield per unit tissue mass and high overall DNA yield. For this experiment, we conducted a total of 123 extractions from tissue samples of nine different masses: 1, 2, 4, 8, 10, 12, 14, 16, and 20 mg. This range was chosen because 1 mg was determined to be the smallest mass that could be reliably manipulated by the experimenter and 20 mg was the maximum mass recommended by our extraction protocol. Tissues were assigned to a sample mass if they were within 0.5 mg of the target mass. In eight cases, there was insufficient tissue to subsample the desired tissue mass and the actual subsample mass was therefore more than 0.5 mg outside the targeted masses. In these instances, tissues were placed in the category to which they were closest, and all were less than 1.2 mg from the target mass. Tissue samples for this experiment were 24 liver tissue samples obtained from Malagasy frogs in 2016 which were all from the family Mantellidae and one sample from Ranidae. Each tissue was sampled 4–12 times at various masses depending on the total tissue mass of the original sample. All of the samples used in this experiment were initially preserved in ethanol and stored at room temperature for a period of several weeks and up to 2 months before being transferred to cryogenic storage in either a mechanical ultracold freezer (−80 °C) or a liquid nitrogen cooled dewar (−180 °C). In each extraction run, four tissues each with four subsamples were extracted for a total of 16 extractions. The data was analyzed using a least squares regression to fit a trend line.

## Experiment 3: Consistency of extraction yield at an optimal mass

Our third experiment assessed the consistency of extraction yield from tissue subsamples at a sample mass identified in Experiment 2 that results in both high DNA yield per unit mass and high overall DNA yield without involving masses so large as to permit only one or two extractions from small tissue samples. The subsample mass that best met these criteria was 8 mg. Because this experiment required four subsamples of 8 mg from each tissue, large samples such as those from Mantellidae were needed. Six Mantellidae tissues were sampled for a total of 32 subsamples (two tissues were used twice due to a lack of suitable tissues). In each extraction run, four tissues each with four subsamples were extracted for a total of 16 extractions.

**Experiment 4: Impact of age on extractions using the optimal mass**

The fourth experiment was conducted using 44 historical anuran samples including both ethanol preserved and flash frozen samples. These samples belonged to several different frog families: Bufonidae (three samples), Dendrobatidae (10), Hylidae (17), Leptodactylidae (11), and three from unknown families. These tissues ranged in collection date from 1984 to 2001 and included both liver and muscle tissue. We sampled, extracted, and quantified 8 mg of each tissue using the same procedure as described above. Data was analyzed using a Pearson's correlation test.

## RESULTS

### Experiment 1: Testing the effectiveness of the "eyeball" method for obtaining tissues appropriate for extraction

We found that coarse visual estimates of tissue subsamples resulted in a wide range of resulting tissue masses (0.65–14.93 mg). The mean mass was 3.33 mg with a standard deviation of 3.32 mg. All but the smallest of the tissues extracted during this experiment resulted in DNA yields that exceeded our 1,000 ng threshold. We also found that DNA yield is significantly positively correlated with original tissue mass (Pearson correlation test: $t = 5.2299$, $r = 0.7600$, df = 20, $p$-value < 0.001, Fig. 1). Additionally, when the three samples with the greatest mass were removed from the analysis, DNA yield was still found to be significantly positively correlated with tissue mass (Pearson correlation test: $t = 2.3112$, $r = 0.4890$, df = 17, $p$-value = 0.0336, Fig. 1).

### Experiment 2: Identification of optimal tissue mass for effective and efficient extraction

In the second experiment, we recovered a non-linear relationship between tissue mass and both concentration and total DNA yield (Fig. 2). The smallest tissue subsamples (1, 2, and 4 mg) yielded a mean of 76.8 ng/µL of DNA. The intermediate tissues (8, 10, and 12 mg) yielded a mean of 123.5 ng/µL of DNA. The largest tissues (14, 16, and 20 mg) yielded a mean of 144.6 ng/µL of DNA. These data were best fit by the natural log equation $y = 3,317.2 * \ln(x) + 5,030.3$ ($R^2 = 0.29$, $p$-value < 0.0001). The relationship between tissue mass and DNA concentration shows a gradual decrease in the DNA gained per mg of tissue as the total tissue mass increases. While the natural log function does not have an asymptote, it may reach a point where the extra DNA that could be obtained is so little that it is not worth the additional destructive use of limited tissue resources. The intermediate and large tissue masses (8 mg and higher) also tend to result in higher overall DNA yields. Although these masses tend to result in both higher DNA concentrations and higher overall DNA yields, yield per unit mass is greatest for the small tissues, with a mean of 3,444.5 ng DNA/mg tissue, as compared to 1,288.7 ng DNA/mg tissue for intermediate masses and 922.6 ng DNA/mg tissue for large masses. These data were best fit by the natural log equation $y = -1,337 * \ln(x) + 4,470.3$ ($R^2 = 0.55$, $p$-value < 0.0001).

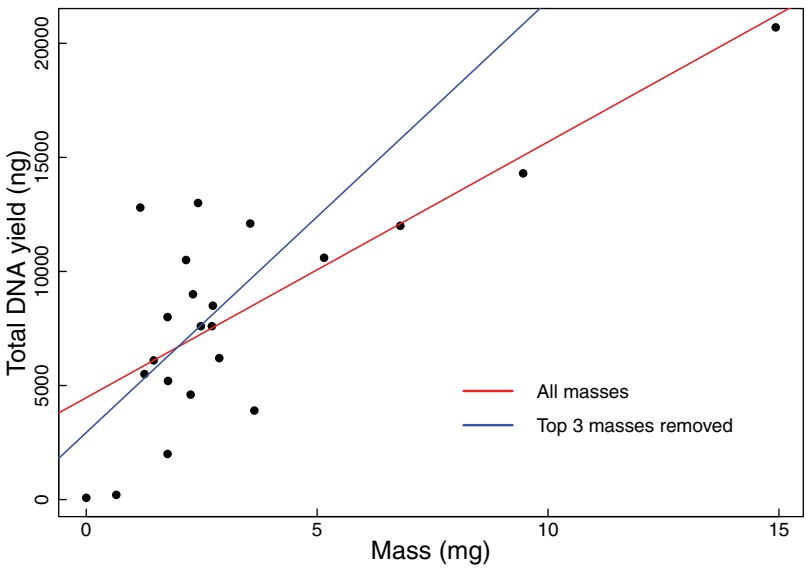

**Figure 1 Total DNA yield vs tissue mass in Experiment 1.** Total DNA yield was positively correlated with tissue mass in Experiment 1. This experiment was designed to test the effectiveness of the "eyeball" method for obtaining tissues appropriate for extraction. Each point represents a single tissue subsample taken in Experiment 1. Tissues were sampled via coarse visual estimate, then massed. We used a Pearson's correlation test to find the relationship between sample mass and total DNA yield. The red line indicates the results of this test when all data points were considered ($t = 5.2299$, $r = 0.7600$, df = 20, $p$-value < 0.001), while the blue line indicates the results when the three greatest masses were excluded ($t = 2.3112$, $r = 0.4890$, df = 17, $p$-value = 0.0336).

## Experiment 3: Consistency of extraction yield at optimal mass

The third experiment further analyzed the precision of using 8 mg of tissue. We analyzed 28 mantellid tissue samples over 32 extractions. One tissue and its four corresponding subsamples were discarded from this analysis resulting in DNA concentrations that were significantly lower from those for all other tissues (Tukey Honest Significant Differences, $p$-values 2.07E−7 to 2.96E−2); we suspect that this tissue was degraded and does not contain sufficient quantities of DNA to result in useful yields following standard DNA extraction methods. The mean DNA concentration from samples extracted during this experiment was 133.75 ng/μL with a mean yield of 13,375 ng of DNA. The mean standard deviation of DNA concentration among subsamples of the same tissue was 19.12 ng and the mean range was 41.86 ng.

## Experiment 4: Impact of age on extractions using the optimal mass

The fourth experiment tested whether the age of tissue samples impacts the expected relationship between sample mass and DNA yield for 44 archival tissues. The average mass of tissue used in this experiment was 7.86 mg with an average yield of 104.56 ng/μL of DNA. This experiment found no correlation (Pearson correlation: $r = -0.06$, $p$-value = 0.6904) between the age of a tissue sample and the concentration of DNA yielded (Fig. 3).

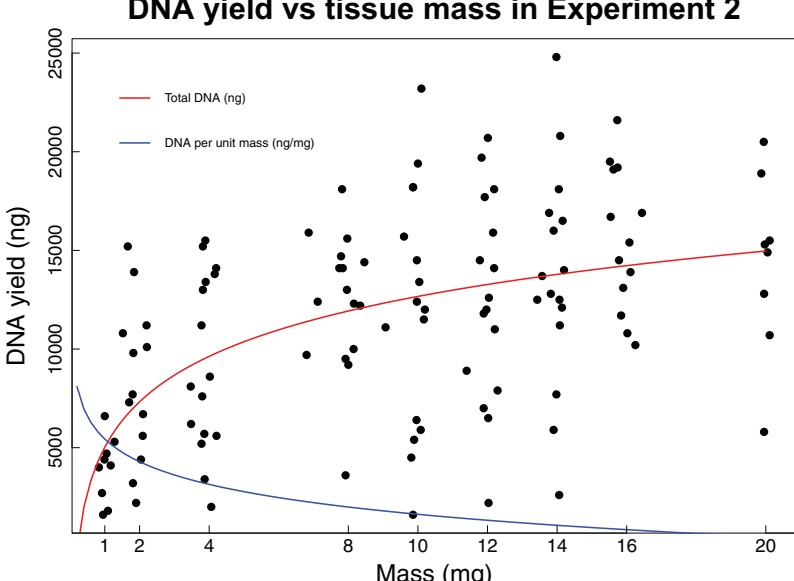

**DNA yield vs tissue mass in Experiment 2**

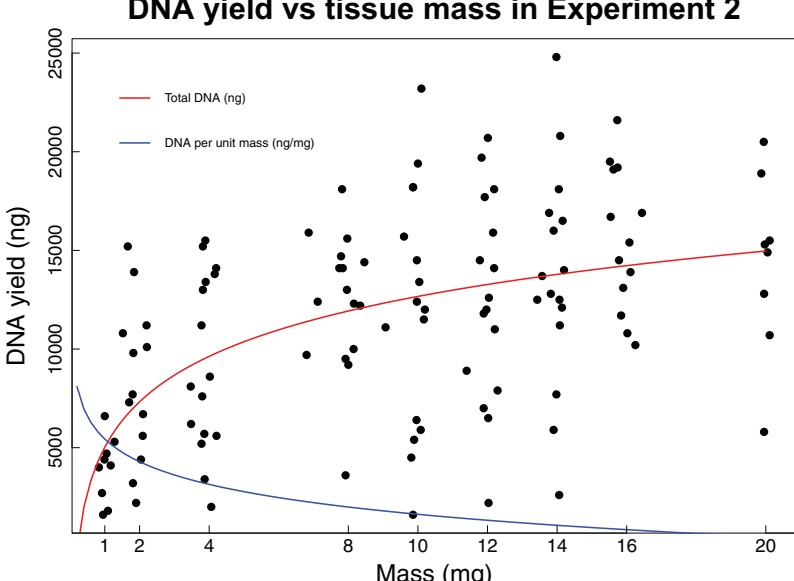

**Figure 2 DNA yield vs tissue mass in Experiment 2.** DNA yield was logarithmically correlated with tissue mass in Experiment 2. This experiment was designed to identify an optimal tissue mass for effective and efficient extraction. Each data point represents an individual tissue subsample taken at one of nine target masses: 1, 2, 4, 8, 10, 12, 16, or 20 mg. The red line shows the trend in total DNA yield across the various masses tested ($y = 3{,}317.2 * \ln(x) + 5{,}030.3$, $R^2 = 0.29$, $p$-value $< 0.0001$), while the blue line shows the trend in DNA yield per unit mass ($y = -1{,}337 * \ln(x) + 4{,}470.3$, $R^2 = 0.55$, $p$-value $< 0.0001$). Trendlines were calculated using a least squares regression.

**DNA yield vs tissue age in Experiment 4**

**Figure 3 DNA yield vs tissue age in Experiment 4.** DNA yield was not correlated with tissue age in Experiment 4. This experiment was designed to test the impact of tissue age on extractions when using an optimal mass. Each data point represents an individual tissue subsample of approximately 8 mg. The red line shows the relationship between the year the tissue was collected and the total DNA yield as calculated using a Pearson's correlation test, $r = -0.06$, $p$-value $= 0.6904$.

## DISCUSSION

The goal of our study was to develop guidelines for sustainable use of tissue samples archived in biodiversity collections that are destructively subsampled for DNA extraction. We found that current tissue sampling methods involving coarse visual assessment of tissue size generally yield sufficient DNA for modern downstream applications. However, the actual yield from samples obtained via the "eyeball" method is highly variable, and, because tissue mass is correlated with DNA yield, massing tissues prior to extraction will increase consistency and efficiency. Intermediate and large tissue masses yielded comparable concentrations of DNA, but small tissue masses had the greatest DNA yield per unit mass. Additionally, sample age was not correlated with DNA yield.

In our first experiment, we showed that the methods currently used by many biodiversity archives, which involve coarse visual estimates of tissue amounts that are considered sufficient for DNA extraction based on prior experience, generally yield more than enough DNA for most modern downstream applications, including whole genome sequencing. However, we also found that tissues subsampled in this manner do not produce consistent amounts of DNA because they encompassed a wide range of masses (0.64–14.93 mg), and DNA yield is strongly correlated with mass. Overall this experiment suggests that use of archived tissue samples would be more efficient if tissues were massed prior to distribution. Of course, this strategy does not come without costs. First, quantification of tissue subsample mass requires a significant additional investment in handling time and access to an expensive analytical balance capable of accurately weighing samples in the 1–20 mg range. As with any increase in handling time, this approach may also result in accelerated degradation of archived samples. However, the benefits of standardization may outweigh these costs, particularly in the case of samples that are only available in limited quantities.

Generally speaking, standardization of tissue masses provided to researchers for extraction will improve the process of intercollection tissue loans because loanees will be sure to receive a quantity of tissue that will result in the required quantity of DNA. The need for an overall standard tissue loan procedure has been previously highlighted (*Droege et al., 2014*) and we believe that, given the strong correlation between tissue mass and DNA yield, standardization of tissue mass could be one important step in this direction. Given the varying specimens housed in different tissue collections, researchers often require tissue loans from other institutions in order to complete their work. It is expected that these tissues will yield sufficient DNA for experimentation, but often collections do not wish to part with the last pieces of a tissue sample. A survey of 45 institutions with genetic resource holdings revealed that none of the 93% of institutions that offered loans sent loanees the entire tissue sample, and amount of tissue sent varied between institutions (*Zimkus & Ford, 2014*). For example, 25% of collections reported sending enough tissue for two extractions and 9% sent enough for three extractions, but only 21% of institutions quantified tissue sent (either by volume or mass). The loan procedures posted on the websites of seven major herpetological collections in the United States (Berkeley Museum of Vertebrate Zoology, California Academy of Sciences, Museum

of Comparative Zoology at Harvard University, Smithsonian Museum of Natural History, University of Florida, University of Kansas, and University of Texas) revealed that these collections provided detailed and well-defined loan procedures for whole animal specimens, but generally provide little detail on procedures for providing genetic resources. Correspondence with collections managers at these institutions revealed a variety of approaches and techniques for determining the amount of tissue to provide researchers requesting access to genetic resources, including qualitative visual assessment, tissue volume, the minimum tissue required for the proposed project, and approximate mass (C. Huddleston, L. Scheinberg, C. Spencer, B. Zimkus, 2019, personal communications). Standardization of tissue masses would allow loanees to receive a previously agreed upon tissue mass that has been shown to yield appropriate amounts of DNA for their proposed downstream applications, while loaners can improve sustainable use of their tissue collections by only loaning the required amount of tissue.

In our second experiment, we recovered a non-linear increase in DNA concentration and total yield with increasing tissue mass, with the smallest masses resulting in considerably lower concentrations and yields than intermediate or large tissue masses. However, the yield per starting quantity mass of tissue, a measure of how efficiently we are recovering DNA from the original tissue sample, is highest at the smallest masses and declines dramatically with tissue sizes greater than 2 mg. For this reason, the decision about which mass is optimal for extraction will depend on a range of factors including the desired application and the total amount of tissue available. For samples available in only very limited quantities, extractions using only 2 mg of tissue will often be ideal because they generally result in sufficient DNA for most downstream sequencing applications while optimizing efficient use of the available material by maximizing DNA yield per unit tissue used (Fig. 2). In cases where larger initial tissue samples are available, it may be preferable to use somewhat larger tissue masses for extraction because masses of 8 g and larger tend to produce considerably higher DNA concentrations and overall yields than small starting masses. In most cases, a single extraction of a larger tissue that produces somewhat lower yields per unit tissue mass than smaller masses will generally be preferable to repeated extractions of smaller samples due to the significant increases in handling time and other expenses associated with extraction. We recommend subsampling more than 2 mg of tissue when removing samples from biodiversity archives for DNA extraction, depending on the amount of material available. Of course, the optimal tissue mass for DNA extraction will depend on the extraction method being utilized and also the intended downstream applications. For this reason, our results are specific to use of the Promega Maxwell platform. Additional work is required to determine the optimal tissue mass to subsample when other extraction methods are being employed. However, it is likely that all these methods will exhibit increased concentration and yield with tissue masses that are larger than the minimum that can be manipulated.

Our fourth experiment suggests that concentration and yield from samples obtained over a 25-year interval are not significantly correlated with age, reflecting previous findings that extraction quality is not correlated with age (Sawyer et al., 2012;

*Choi, Lee & Shipunov, 2015*). This suggests that the same masses identified as being ideal for extraction of recent samples are also appropriate for historical samples. However, we did not evaluate other important factors influenced by age such as fragmentation, which might have similar yields with increasing age, but higher fragmentation.

We primarily focused on tissue types that are most commonly housed in biodiversity archives and used for extractions. We therefore did not analyze several other sources of genetic material in natural history collections, namely formalin-fixed specimens and tissue samples treated with RNAlater. Previous work has attempted to extract high quality DNA from formalin-fixed specimens with varying results (*Hykin, Bi & McGuire, 2015*; *Jaksch et al., 2016*). Because the procedures used for these types of extractions are more involved and less often used, we chose not to include any formalin-fixed tissue subsamples in our study but recommend repeating our study with these specimens once extractions procedures are better developed. Conversely, specimens treated with RNAlater are often deliberately collected fresh from the field for a specific hypothesis. These specimens are often used for RNA-Seq applications to assess variation in gene expression in different tissue types (*Wang, Gerstein & Snyder, 2009*), but are increasingly used for DNA work as well. These extractions were not included in this study due to the lack of these tissues in the University of Kansas herpetological collection and also because RNA extraction protocols have many more variables to consider (e.g., time to freezing, freezing temperature, amount of RNAlater used, freeze-thaw cycles). Further research is needed to determine if the results of our study also apply to RNAlater treated tissue samples.

## CONCLUSIONS

Our experiments analyzed current practices in tissue subsampling and DNA extraction in biodiversity collections. We found that extractions using 2–8 mg of tissue were the most efficient and did not recover a strong correlation between DNA yield and tissue age. Two specific recommendations for improving sustainable use of genetic resources in biodiversity archives emerge from our study. Our first recommendation could be achieved with relatively minor adjustments to existing loan procedures while the second would require a dramatic change in how biodiversity archives provide researchers with access to genetic resources.

First, we discussed in detail the potential value of providing researchers with tissue samples of known mass. By standardizing the mass of tissues provided as gifts to researchers, the loaning institution be will be better able to ensure that researchers are provided with sufficient material while also being able to make more informed decisions about how limited resources are destructively sampled.

Our second recommendation derives from our finding that even very small quantities of tissue often produce far more DNA than is required for most applications. For example, we found that tissues subsamples weighing 8 mg tend to yield more than 13 times the amount of DNA that is required even for whole genome shotgun sequencing. In most cases, excess DNA obtained by researchers who receive tissue loans is discarded. Even in cases where institutions are capable of archiving extracted DNA and request return of

unused material, this rarely happens in practice because it is very difficult to enforce such requests. As a result, the current practice of providing researchers with even very small tissue samples from permanently archived material for use in individual sequencing projects results in highly non-optimal use of limited archived resources. In the case of the University of Kansas herpetological collections, we are increasingly finding that popular tissue samples have been nearly or completely exhausted after providing multiple prior tissue gifts to researchers. In many cases, these researchers sequenced only one or a few loci via Sanger sequencing, meaning that we provided them with orders of magnitudes more irreplaceable genetic material than was necessary for their work.

One possible solution to this extremely inefficient use of archived resources is to end the practice of providing researchers directly with subsamples of archived tissues and to instead provide researchers with only the amount of extracted DNA that is required for their particular application. For example, in the case of a project involving Sanger sequencing of one or two loci, a biodiversity archive could send the researchers 50–100 ng of extracted DNA instead of a destructively subsampled piece of tissue that is expected to yield 10,000 ng of DNA. Rather than resulting in researchers discarding large quantities of irreplaceable DNA, this practice would lead to archiving this material so that it could then fulfill subsequent requests for genetic material from the same specimen. However, this would require DNA extraction by biodiversity archive staff followed by quantification and provision of the appropriate amount of DNA for the researcher's required application. It would also require biodiversity collections to develop archival collections of not only tissues, but also extracted genomic DNA.

Although this approach could result in considerably more sustainable use of limited tissue resources, it does not come without substantial costs. First, it would require that staff at biodiversity collections extract and quantify DNA rather than merely sending a tissue sample. In many cases the staff responsible for preparing tissue loans will not have the requisite expertise, access to the necessary laboratory facilities, or time. Second, in-house extraction would require new protocols and facilities for archiving extracted DNA. Whether these costs are worthwhile will depend on the amount of material available and how heavily it is used by the research community. In the case of the University of Kansas herpetological collections, we now provide researchers only with an amount of extracted genomic DNA required for their research because we are finding that a significant number of samples in our archive have been used to the point that little or no tissue remains. We recommend that other biodiversity collections experiencing such over-use consider adopting a similar approach because it will radically improve sustainable use of genetic resources.

## ACKNOWLEDGEMENTS

We thank Luke Welton for his assistance in accessing the University of Kansas herpetological collections. We thank the following curators and collections managers for their personal communications: Chris Huddleston, Lauren Scheinberg, Carol Spencer, and Breda Zimkus. We thank Shea Lambert for his participation in Experiment 1.

### Funding

Funding for this project provided by National Science Foundation award 1457774 to Richard E. Glor. There was no additional external funding received for this study. The funders had no role in study design, data collection and analysis, decision to publish, or preparation of the manuscript.

### Grant Disclosures

The following grant information was disclosed by the authors:
National Science Foundation award: 1457774.

### Competing Interests

The authors declare that they have no competing interests.

### Author Contributions

- E. J. Tuschhoff conceived and designed the experiments, performed the experiments, analyzed the data, prepared figures and/or tables, authored or reviewed drafts of the paper, and approved the final draft.
- Carl R. Hutter authored or reviewed drafts of the paper, provided laboratory training, and approved the final draft.
- Richard E. Glor conceived and designed the experiments, authored or reviewed drafts of the paper, and approved the final draft.

### Data Availability

The raw data on tissue samples—including specimen numbers, year collected, family, experiment/trial in which they were used, date extracted, tissue class (for experiment 2), tissue type, mass, and DNA extracted—is available in File S1. The specimens are housed in the herpetological collection at the University of Kansas Biodiversity Institute. The R code used to run analyses is available in File S2.

### Supplemental Information

Supplemental information for this article can be found online at http://dx.doi.org/10.7717/peerj.8369#supplemental-information.

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
