# Peer review of "Improving sustainable use of genetic resources in biodiversity archives"

_PeerJ, doi:10.7717/peerj.8369_

## Round 0.1 · original submission · Minor Revisions

Two reviewers have found this manuscript to contain several experiments that were generally well-performed and which will provide important information for the future maintenance of genetic resources in natural history collections. I agree with the reviewers feedback and recommendations. Both reviewers made several comments that will improve this manuscript and I encourage you to address each of these recommendations carefully.

Reviewer 1 makes an important point regarding RNAlater and I think the authors would do well to discuss tissue collection with this medium given it is increasingly common to use. It's also not clear necessarily how DNA yield may differ between ethanol-preserved versus RNAlater preserved tissues and is something worth engaging your readership with.

I have one additional main comment that reflects Reviewer 2's comment with respect to the "strong" correlation between mass and total DNA yield in Experiment 1. I've never used a Promega fluorometer to measure DNA yield but am a little surprised by the amount of variation in DNA yield as a function of tissue mass (Figure 2). I haven't thought much before about variation in DNA yield as a function of mass but given almost all of the samples in Experiment 2 were from a single frog family collected from Madagascar in the same year and were of the same tissue type, I assumed there would be a tighter relationship between mass and yield for this experiment. You go on in the discussion to state that DNA yield is strongly associated with mass; while there certainly is an association from both Experiment 1 and 2, I'm not sure it's so strong that more precise massing of tissue prior to extraction is going to have a large effect on yield relative to other issues like tissue degradation. I think the authors would do well to discuss this variation in the data more explicitly, what might drive this variation, and whether your recommendations for massing tissue samples is indeed worth it given the spread of the data.

I have an additional minor comment on line 166: please indicate who the "two experienced scientists" were who subsampled tissues.

I look forward to seeing the revised version of this manuscript.

·

Basic reporting

Overall, this is an excellent, concise manuscript with clear objectives and a logical flow throughout. The main questions asked by the authors are of relevance to any researcher working not only with museum collections but performing DNA extractions from any tissue.

I have a few minor comments pertaining to figures, one suggestion for an added citation, and a handful of comments relating to spelling/grammar.

Lines 97-98, the authors state that “previous studies have not tested the correlation between age and extraction success…”; there is a reference that should be cited here:
Chambers & Hebert. 2016. Assessing DNA Barcodes for Species Identification in North American Reptiles and Amphibians in Natural History Collections. PLoS ONE https://doi.org/10.1371/journal.pone.0154363
While Chambers & Hebert (2016) examined tissue age vs DNA barcode sequence length rather than extraction success, I think it is still relevant to include their results in this manuscript.
In general, all the figure captions need to provide more detail. Also, I suggest removing the titles from the actual figure images. For all figures, perhaps instead of the figures mentioning the experiments only (e.g., “Total DNA yield vs tissue mass in Experiment 1”), there could be some additional descriptive text that allows the figures (paired with the captions) to stand alone, without the reader heaving read the text.

For the Figure 1 caption, when the authors state “the red line indicates the correlation…”, it would be helpful to also state the statistical method that was used to determine this correlation (Pearson’s correlation), and also relevant results (R2 value, p-value, etc.). The same applies to the methodology represented in Figs. 2 and 3.

Minor grammatical/formatting comments:
ll. 71-72, 74-75, 79, 344: References not in chronological order.
ll. 92: change “sample” to “samples”.
ll. 93: change “for use by molecular” to “for use in molecular”.
ll. 233: Missing closing parenthesis.
ll. 245: Should be “its”, not “it’s”.
ll. 367: Change “shot-gun” to “shotgun” to stay consistent with the rest of the ms.
ll. 369: Add comma following “material”.
ll. 379: Change “to end to the” to “to end the”.
ll. 387: Change “This solution however,”; awkward.

Experimental design

I appreciated that the authors maintained consistency throughout the ms through the use of four experiments; this made it easy to keep track of and also the experimental design was thoughtfully laid out and was easy to follow logically to complete goals.

A few minor comments:
Lines 127, 129: How was “complete tissue digestion” or “level of tissue digestion” quantified?
Lines 149-150: Why was elution buffer volume doubled, contrary to kit recommendations?
Lines 160-161: It would be helpful to have a citation here for the amount of DNA recommended to perform genome sequencing.

Validity of the findings

I have no comments for improvement of the discussion; it is well laid out and easy to follow. One suggestion that I think would be of interest to researchers is a short comment by the authors on the utility of RNAlater for preserving tissues in the field; this seems to be becoming more common practice among researchers, and a comment in the discussion (perhaps as a future direction of research) might be worthwhile.

Reviewer 2 ·

Basic reporting

No comment, see general comments for my review

Experimental design

No comment, see general comments for my review

Validity of the findings

No comment, see general comments for my review

Additional comments

My entire review is contained in these general comments.

This manuscript addresses the quantity and quality of DNA coming from frozen herpetological tissues. The purpose is to create some benchmarks for how much tissue is needed to get sufficient DNA in a world of finite tissue resources. This manuscript fills an interesting vacant niche in the literature, in the sense that most of the articles now appearing on this subject address DNA from historical museum specimens. Not many people are thinking much at the moment about preserving the DNA actually in frozen tissue collections (aside from tissue collection managers!). But it is an important issue worthy of highlighting, and the results are interesting and in some cases counter-intuitive, especially:

• The larger the tissue mass, the more decreasing returns of DNA per tissue mass
• DNA amount does not decrease with length of time in the freezer

I think the authors could do a bit more to place the topic in the broader context of museum collections and ancient DNA, and as it stands the literature cited is a bit thin, so they could definitely expand it out. For instance, if tissue runs out, why can’t we just use bits of the specimen? Well, for herps there is the issue of formalin preservation, which is addressed in recent publications:

Hykin, S.M., Bi, K. and McGuire, J.A., 2015. Fixing formalin: a method to recover genomic-scale DNA sequence data from formalin-fixed museum specimens using high-throughput sequencing. PLoS One 10: p.e0141579.

Ruane, S. and Austin, C.C., 2017. Phylogenomics using formalin‐fixed and 100+ year‐old intractable natural history specimens. Molecular Ecology Resources, 17: 1003-1008.

In terms of experimental design, the study is solid. On the one hand the experimental design is quite thorough and the authors have large sample sizes and address four separate experiments:

1. Does eyeballing tissue size give you enough DNA for most uses?
2. What is the best and smallest mass to get you there?
3. How consistent is DNA yield from specific tissue sizes?
4. Does age of the frozen tissue matter for DNA yields?

There are a few weak spots in their experimental design, but I do not think they are fatal to the study. For example, as the authors acknowledge, they use an extraction method that is not very common. But they do this in service of getting very standardized quantities that minimize human error, and there is no reason to think the results will not be generalizable to other, more common extraction methods.

Another small issue with Experiment 1 is that, because they combined two experiments within one (does the eyeball method produce enough DNA and is mass correlated with DNA), their masses ended up clumped around middle values, so a big part of the correlation is just driven by the four points on the higher end of the mass spectrum. Again, not fatal, but worthy of mentioning.

Ln 243-251 As for findings, I did not feel the results section for experiment 3 answered the question posed. Only means are reported, not consistency measures.

In the conclusions, the authors could significantly tighten up the writing as there is much redundancy within the conclusions and with other parts of the Discussion.

Overall this was a nice, solid study.

Minor comments:

ln 245 its
Ln 264 remove “while”
Ln 291 remove one of the “that”'s

---

## Round 0.2 · Minor Revisions

Thank you for your careful attention to the reviewers' comments - this is a much improved manuscript. I have only two small revisions I believe should be made before this can be accepted.

One pertains to the citations for species declines in the Introduction. Specifically, the Scheele citation is incorrect (it should be Scheele et al. 2019). However, I recommend the authors find a more suitable citation. A response to this paper is in press right now that demonstrates the study cannot be replicated. Given this publication does not address amphibians declines generally (only in relationship to chytrid) and issues with it being replicated, it might be best to change this. Additionally, there are perhaps more suitable citations for the point that sentence is making anyway than either this citation or the Pyron and Jetz citation. These include inclusion of a citation on the Nagoya Protocol which is relevant to the additional text you added to this sentence.

I recommend the following citations:

Dirzo et al. 2014. Defaunation in the Anthropocene. Science 345: 401-406.

Roll et al. 2017. The global distribution of tetrapods reveals a need for targeted reptile conservation. Nature Ecology & Evolution 1: 1677-1682.

Stuart et al. 2004. Status and trends of amphibian declines and extinctions worldwide. Science.

Watanabe. 2015. The Nagoya Protocol on Access and Benefit Sharing. BioScience.

Additionally, your new text in the Discussion implies that tissues preserved in RNAlater are intended only for RNA research. This is certainly not true. It is now common practice for many researchers and museum professionals to preserve tissues as often as possible in RNAlater even if only using tissues for DNA extractions. Preservation in RNAlater can increase DNA yield and so is increasingly preferred. Please modify this section accordingly given RNAlater is and will continue to be an important medium for preserving tissues for DNA work.

I look forward to receiving the revised version of your manuscript.

---

## Round 0.3 · accepted · Accept

Your revised manuscript is greatly improved and presents interesting and useful information for a wide range of biologists. I believe this will help facilitate an important conversation about our use of museum collections.